# vvv2_align_SE, vvv2_align_PE/vvv2_display: Galaxy-Based Workflows and Tool Designed to Perform, Summarize and Visualize Variant Calling and Annotation in Viral Genome Assemblies

**DOI:** 10.3390/v17101385

**Published:** 2025-10-17

**Authors:** Alexandre Flageul, Edouard Hirchaud, Céline Courtillon, Flora Carnet, Paul Brown, Béatrice Grasland, Fabrice Touzain

**Affiliations:** 1Bio Chene Vert, ZI de Bellevue, rue Blaise Pascal, 35220 Châteaubourg, France; a.flageul@labofarm.com; 2ANSES, VIPAC Unit, Laboratoire de Ploufragan-Plouzané-Niort, 22440 Ploufragan, France; celine.courtillon@anses.fr (C.C.); beatrice.grasland@anses.fr (B.G.); 3ANSES, VIRPIG Unit, Laboratoire de Ploufragan-Plouzané-Niort, 22440 Ploufragan, France; edouard.hirchaud@anses.fr (E.H.);

**Keywords:** virus, variant calling, display, bioinformatics, workflow

## Abstract

**Background**: Next-generation sequencing (NGS) analysis of viral samples generates results dispersed across multiple files—genome assembly, variant calling, and functional annotations—making integrated interpretation challenging. Variants often yield numerous low-frequency or non-significant variants, yet only a small fraction are biologically relevant. Virologists must manually sift through extensive data to identify meaningful mutations, a time-consuming and error-prone process. To address these practical challenges, we developed vvv2_display, a dedicated summarization and visualization tool, integrated within comprehensive Galaxy workflows. **Results**: vvv2_display streamlines variant interpretation by consolidating key results into two concise and interoperable outputs. The first output is a PNG image showing alignment coverage depth and genomic annotations, with significant variants displayed along the genome as symbols whose height reflects frequency and shape indicates the affected protein. At a glance, this enables virologists to identify all deviations from a reference viral genome. Each significant variant is assigned a unique identifier that directly links to the second output: a tab-separated (TSV) text file listing only high-confidence variants, with frequencies, flanking nucleotides, and impacted genes and proteins. This cross-referenced design supports rapid, accurate, and intuitive data exploration. **Availability**: vvv2_display is open source, available on Github and installable via Mamba.

## 1. Introduction

Next-generation sequencing (NGS) of viral samples typically begins with the assembly of viral reads to reconstruct the complete or partial genome. This is followed by functional annotation and variant calling—key steps for understanding viral evolution, host adaptation, and pathogenicity. Each of these stages generates large, complex text-based outputs that require extensive manual curation to extract biologically meaningful insights. For virologists, synthesizing this information into a coherent interpretation is both time-consuming and prone to error, especially when dealing with low-frequency or functionally relevant variants.

While several bioinformatics workflows offer solutions for summarizing and visualizing variant data, most are tailored to high-priority pathogens with well-curated reference databases—particularly SARS-CoV-2 [1], and, to a lesser extent, mpox, influenza A virus, and respiratory syncytial virus [2]. These specialized tools often rely on pathogen-specific annotations, making them unsuitable for emerging, rare, or poorly characterized viruses lacking dedicated resources.

To address this need and accelerate virological analysis, the VVV tool [3] was previously developed to integrate identification, de novo assembly, annotation transfer, and variant calling within a Galaxy [4,5] environment. While VVV proved useful in practice, it was not originally designed for Galaxy’s framework and operates as a monolithic workflow without intermediate feedback. As a result, it provides no user-facing error messages during execution, making troubleshooting difficult and limiting transparency.

To improve usability, robustness, and interpretability, we have completely re-engineered the downstream analysis pipeline with vvv2_display—a modular, Galaxy-native tool dedicated to the graphical summarization and visualization of variant calling results, which constituted the main added value of the original VVV. Furthermore, by developing a dedicated Galaxy wrapper for vvv2_display, we enabled its integration into two new end-to-end workflows: vvv2_align_SE and vvv2_align_PE. These workflows perform, in a transparent and user-friendly manner, all upstream steps—including read quality control, removal of PCR duplicates, reference-based alignment, annotation, and variant calling—culminating in the automated generation of vvv2_display’s concise and interpretable outputs.

This new suite offers a general-purpose, database-agnostic solution for viral variant analysis, designed for broad applicability across diverse viruses—even those without curated reference databases—while ensuring full reproducibility, error reporting, and intuitive result interpretation within the Galaxy platform.

## 2. Materials and Methods

### 2.1. vvv2_display: Visualization and Summarization of Variant Data

vvv2_display (version 0.2.4.0), the core tool presented here and summarized in Appendix A, handles the downstream processing and visualization of results generated by the updated pipeline. Built as a Conda/Mamba-installable package, it consists of Python 3.9 and R 3.4.3/ggplot2 3.5.2 scripts that integrate

-Significant variants from VarDict 1.8.3 (bed file);-Functional annotations from VADR 1.6.4 (passed and failed annotations tsv file, seqstat text file);-Genome-wide coverage depth from samtools depth 1.15.1 (text file).

The output is condensed into two intuitive files:

1-A PNG image displaying two vertically aligned panels (see Figure 1a):
-The top panel shows coverage depth along the genome (logarithmic scale optional).-The bottom panel displays gene annotations as staggered rectangles (traditional representation of often overlapping viral genes) and a continuous gene-colored line, with a legend for clarity.-Significant variants are plotted as symbols along the *x*-axis (genomic position), with vertical position indicating variant frequency (0% at bottom, 100% at top).-The symbol shape encodes the affected protein, enabling immediate functional interpretation.-Each variant is labeled with a unique number, assigned in increasing order by genomic position.
2-A tab-separated (TSV) text file that describes each labeled variant with 10 columns (see Figure 1b):
-position: genomic position;-ref: reference nucleotide (s);-alt: variant nucleotide (s);-freq: variant frequency;-gene: associated gene (s);-prot: affected protein (s);-lseq: flanking sequence (left);-rseq: flanking sequence (right);-isHomo: Boolean flag indicating whether the variant lies within a homopolymer region.

The is Homo flag is particularly important for data generated on Ion Torrent or Nanopore platforms, which are prone to false indels in homopolymer tracts. Such artifacts may cause frameshifts in coding regions and lead to incorrect protein predictions; this flag helps users critically assess potential technical errors.

### 2.2. Installation

As a conda package, vvv2_display can be installed using the following commands (assuming mamba is installed):# creation of the mamba environment and installation;mamba install –name vvv2_display –c bioconda vvv2_display –y;# environment activation to use the tool;mamba activate vvv2_display;# shows the man page for the tool;vvv2_display.py -h.

### 2.3. Integration and Workflow Automation

Thanks to a dedicated Galaxy wrapper (available on the Galaxy ToolShed), vvv2_display can be seamlessly integrated into any Galaxy workflow. It is the final step in two new end-to-end workflows: vvv2_align_SE (single-end) and vvv2_align_PE (paired-end), which automate quality control, PCR duplicate removal, alignment, annotation, variant calling, and summarization. Appendix A shows the vvv2_align_PE_bwamem2 Galaxy workflow for paired-end Illumina data, adding two consensus outputs: strict (major nucleotides only) and IUPAC (ambiguous bases only at variant sites)—to the standard vvv2_display results.

**Figure 1 viruses-17-01385-f001:**
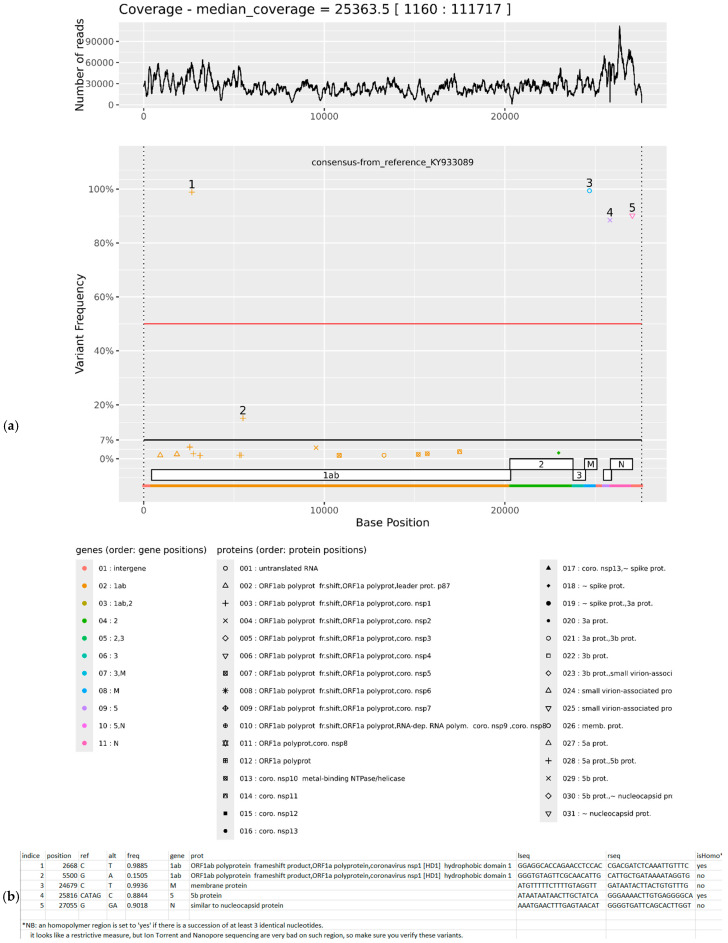
Overview of VVV2_display outputs in Galaxy for a sample infected with infectious bronchitis virus. (**a**) Proportions of variants across the viral genome and annotated genes. (**b**) Text file summarizing each significant variant, including its genomic position, nucleotide change, and associated gene.

## 3. Results

### 3.1. Evaluation of vvv2_display

vvv2_display was evaluated using each of the VADR annotation models across various combinations of reference genomes (or assemblies) and their corresponding read datasets (see Table 1).

Accurate annotations were consistently obtained for all viral models processed by VADR. Both de novo assemblies and reference genome annotations for coronaviruses were in full agreement with the genomic features documented in the corresponding GenBank records (11/11 genes). For flavivirus, VADR correctly identified the position of the NS5 protein—a hallmark feature—even though this annotation was not explicitly reported in the reference GenBank file (1 gene, 15/14 proteins). All other gene and protein positions and names were accurately annotated. In the case of norovirus, the GenBank file included additional features (*ORF4* gene, VF1 protein, 3/4 genes) that had been manually added by the NCBI curation team. All genes of porcin circovirus 2 were accurately predicted (3/3 genes).

A dedicated VADR model for Porcine Circovirus (PCV) was developed using all complete genomes of PCV types 1, 2, 3, and 4 available in the NCBI nucleotide (nt) database as of 12 July 2023. This model is publicly available for download at Zenodo (https://zenodo.org/records/15065124 (accessed on 19 June 2025)).

It should be noted that VADR appears better suited for viruses with genomes smaller than 30 kb. For instance, the SARS-CoV-2 model—characterized by a genome of approximately 29 kb—required substantial RAM usage (23 GB in addition to baseline system allocation). To limit memory consumption and enable efficient multithreading, the VADR options -glsearch –split –cpu 4 were integrated and fixed within the dedicated Galaxy wrapper. These settings optimize computational resource utilization while preserving annotation accuracy, ensuring reliable performance on standard computing infrastructure. This standardization guarantees computational feasibility and result comparability across different viral genomes. Furthermore, it enhances the security and stability of the Galaxy instance hosting this workflow, as users typically cannot anticipate the computational demands of their analyses, which may vary significantly depending on viral genome characteristics and sequencing data properties.

To improve adaptability to virus-specific features, two additional options have been implemented in vvv2_display. First, users can display the *y*-axis on a log_10_ scale in the sequencing depth coverage plot, improving visual clarity in cases of high coverage variability. Second, users can define a custom threshold for variant proportion to classify a mutation as significant and retain it in the final summary TSV file. This allows adjustment of the significance threshold according to the expected mutation rate of the virus under study, thereby enhancing the biological relevance and analytical precision of the results. As an indication, for RNA single-stranded positive-sense viruses—including enveloped viruses such as coronaviruses (e.g., SARS-CoV-2, turkey coronavirus), flaviviruses (e.g., dengue virus, Spondweni virus), hepatitis C virus, and nonenveloped caliciviruses (e.g., norovirus, feline calicivirus)—we applied a 7% significance threshold. For single-stranded DNA viruses (e.g., porcine circovirus) and single-stranded RNA negative-sense enveloped viruses (e.g., influenza A virus), we set the threshold to 10%.

Multi-segmented viruses (e.g., Influenza) are processed similarly to multi-contig assemblies. PacBio and Oxford Nanopore sequencing data are supported, provided that the Minimap2 aligner [7] is used, or appropriate options are specified for BWA-MEM2 (e.g., -x ont2d for Nanopore data) [8], as implemented in workflows 1740 and 1741, respectively.

### 3.2. Galaxy Implementation and vvv2_align_SE/vvv2_align_PE Workflows

For end-to-end usability, four Galaxy workflows were developed to support vvv2_display:-vvv2_align_PE_bwamem2 (Illumina paired-end);-vvv2_align_SE_bwamem2 (Illumina single-end/Ion Proton);-vvv2_align_SE_bwamem2_nanopore (Nanopore);-vvv2_align_SE_bwamem2_pacbio (PacBio).

These workflows integrate the following tools:-Fastp 0.23.2-galaxy0 [9] for read quality trimming and adapter removal;-BWA-MEM2 2.2.1+galaxy1 or Minimap2 [7,8] for alignment;-rmdup 2.0.1 [10] to remove PCR duplicates;-VarDict 1.8.3 [11] for variant calling;-VADR 1.6.4 (Galaxy wrapper 0.2.0) [12] for genome annotation;-vvv2_display 0.2.4.0 (this work) for intuitive visualization and summary outputs;-BCFtools 1.15.1+galaxy2/3 [13] for consensus sequence generation.

These fully integrated workflows are publicly available on WorkflowHub at

-https://workflowhub.eu/workflows/1738 (PE Illumina, version 1, accessed on 19 June 2025);-https://workflowhub.eu/workflows/1739 (SE Illumina/Proton, version 1, accessed on 19 June 2025);-https://workflowhub.eu/workflows/1740 (SE Nanopore, version 1, accessed on 19 June 2025);-https://workflowhub.eu/workflows/1741 (SE PacBio with Minimap2, version 1, accessed on 19 June 2025) [7].

As a typical performance benchmark, the complete processing of a SARS-CoV-2 sample (see Table 1) using the Illumina paired-end workflow (1739)—including read cleaning, alignment, annotation, variant calling, and vvv2_display summarization—required less than 3 minutes and less than 13 GB of RAM on a standard Galaxy server (10 threads for samtools, 8 for bwa-mem2, 7 for bcftools, 4 for vadr and fastp, 2 for vardict, 1 for other processes on a Intel Xeon Gold 6226R CPU at 2.90GHz). Each step generates its own log file, facilitating troubleshooting and quality control.

## 4. Discussion

### 4.1. From VVV to vvv2: A Modular Redesign for Improved Usability and Transparency

The original VVV pipeline integrated multiple steps for viral genome analysis—read identification, reference alignment, de novo assembly, annotation transfer, variant calling, and result summarization—into a single Snakemake workflow launched via a unified Galaxy wrapper. While functional, this monolithic design had a critical limitation: errors occurring at any step were not reported individually, as no per-step logs were exposed to the user. This lack of transparency hindered troubleshooting and reduced usability within the Galaxy environment.

Furthermore, VVV relied on legacy components that are no longer optimal for modern NGS analysis. Viral identification was carried out using Megablast 2.7.1 [14] to align reads against a local copy of the NCBI nt database. The alignment outputs were subsequently parsed and converted via a Perl script dependent on one now-deprecated BioPerl library. By September 2025, the nt database had grown beyond 1 TB, with its lightweight counterpart (core_nt, >230 GB) still posing storage and maintenance challenges for routine deployment. Today, more efficient and universal taxonomic assignment tools are available, such as MagicBlast [15] for read-based classification, MegaBLAST [16] for contig alignment, DIAMOND [17] for distant homology detection, or fast k-mer-based methods like Kraken2/Bracken [18], Sylph [19], and Mash Screen [20].

### 4.2. Improved Alignment, Assembly, and Annotation Strategies

In VVV, read alignment and assembly were performed using BWA-MEM v0.7.17-r1188 [21] and SPAdes v3.12 [22], respectively. These have since been superseded or complemented by faster and more versatile tools now integrated into modern Galaxy workflows. BWA-MEM2 [8] and minimap2 [7] offer improved speed and scalability, while assembly can now be tailored to sequencing technology: SPAdes for Illumina data, Canu [23] for long reads (Nanopore or PacBio). Although Flye and viralFlye [24,25] perform well in metagenomic contexts, they are not recommended for viral genomes shorter than 1000 nt—the current minimum required for read overlap—limiting their applicability in targeted viral sequencing.

Functional annotation in VVV relied on Smith-Waterman alignment to a reference genome followed by footprint-based gene assignment. For the updated pipeline, we evaluated VAPID 1.2 [26], which delivers excellent results on well-characterized viruses. However, it performed poorly when applied to the turkey coronavirus assembly, a less-annotated genome. In contrast, VADR 1.6.4 [12] demonstrated robust performance on both known and novel viral sequences. VADR uses Hidden Markov Models (HMMs) to identify and annotate viral features with high accuracy. When a family-specific model is unavailable, users can create a custom HMM following the official tutorial (https://github.com/ncbi/vadr/blob/master/documentation/build.md#top (accessed on 19 June 2025)). To ensure seamless integration into Galaxy, we developed a dedicated Galaxy wrapper for VADR, now publicly available on the Galaxy ToolShed.

Variant calling is performed using VarDict-java 1.8.3 [11], a sensitive tool for detecting low-frequency variants, which already had a Galaxy-compatible wrapper.

### 4.3. Key Improvements over VVV

Compared to the original VVV output, vvv2_display introduces several critical enhancements:-Gene structure visualization using clearly delineated gene boxes, improving readability over VVV’s simple color bands;-Gene color legend for unambiguous gene assignment;-Protein-specific symbols for each variant, enabling direct functional interpretation;-Optional logarithmic scaling of coverage depth for better dynamic range;-User-defined significance threshold for variant filtering.

Table 2 compares the vvv2_display/vvv2_align solutions with existing software.

## 5. Future Developments

In the near future, we plan to implement the following enhancements:-Generation of a VCF file equivalent to the summary TSV, enabling downstream analyses with tools such as SnpEff [27] for functional mutation annotation and Nextclade [28] for clade assignment;-Integration of GenBank file-based annotations, allowing users to retrieve genomic features directly from a reference GenBank file instead of relying solely on VADR predictions—particularly useful when analyzing reference genomes rather than de novo assemblies;-Inclusion of structural RNA elements, such as stem-loop structures and long non-coding RNAs (lncRNAs), either from reference annotations or in silico predictions, to enrich the biological context of the output.

These developments aim to improve interoperability with existing bioinformatics pipelines and provide more comprehensive, functionally relevant genomic insights.

## 6. Conclusions

vvv2_display integrates three user-friendly components into a single, streamlined solution tailored for virologists:-Sequencing coverage depth visualization;-Rapid identification of variants along the genome, including their relative proportions;-Biologically relevant annotations of the genome (reference or assembly).

These features are unified within a single analytical tool designed for seamless integration into a Galaxy instance, enabling efficient, accessible, and reproducible viral genome analysis—even in the absence of dedicated bioinformatics expertise. By combining accuracy, adaptability, and ease of use, vvv2_display supports robust and scalable viral genomics in both research and surveillance contexts.

## Figures and Tables

**Table 1 viruses-17-01385-t001:** Association of vadr model to the reference genome and raw reads data.

Vadr Annotation Model Virus Group (Model Name)	Accession Number of the Reference Genome	SRA Archive Code/Reference of the Reads
SARS-CoV-2 (sarscov2)	MW400961.1	SRR2204988
flavivirus (flavi)	MG182017.2	SRR8133411
coronavirus (corona)	original turkey coronavirus assembly (3 contigs) [6]	raw reads OG30Bp from another organ [6]
calicivirus (calici)	NC_001481.2	SRR16202400
dengue virus (dengue)	MW362474.1	SRR14340780
hepatitis C virus (hcv)	MK139022.1	SRR23502282
norovirus (noro)	NC_008311.1	ERR10977744
porcine circo virus (pcv)	KT719404.1	SRR14333558
avian influenza (flu)	MN254698 (PB2), MN254518 (PB1), OP221387 (PA), OP221382 (HA), MN937705 (NP), OP221383 (NA), OP221384 (MP), MT982385 (NS)	raw reads from the Avian Influenza surveillance program

**Table 2 viruses-17-01385-t002:** Key features compared to existing software.

Software	vvv2_display 0.2.4.0	vvv2_align Version 1	CoVEx Version as of 21 October 2023	VIRUS-MVP v1.0.0
**inputs**	tsv (annot.), bed (variants), [txt (cov depth)]	fastq.gz (reads), fasta (ref, assembly), vadr model name	fastq.gz (reads), fasta (ref), [bed (primer)], [tsv (metadata)]	genome_config.json, gff (annot.), fasta (ref), GenEpiO or GGO (ontology), fasta (assembly), tsv (metadata), tsv (Pokay annot.)
**outputs**	png (variant proportions, annot, [cov depth]), tsv (variants)	png (variants, annot), tsv (variants), fasta (consensi)	html (report), png (prevalence heatmap)	gvf, tsv (funct. annot.), tsv(s) (metadata), pdf (summary)
**interactivity**	no	no	yes	yes
**galaxy integration**	yes	yes	no	no
**status**	stable	stable	stable	stable
**virus groups (dev.)**	9	9	1	2 (2)
**GISAID profiles**	no	no	yes	yes
**displays gene overlaps**	yes	yes	no	no
**scalability to new viruses**	easy, creating vadr model	easy, creating vadr model	no	complex, ontology curation
**handle assemblies**	yes	yes	no	yes
**multi-segment support**	yes	yes	no	Yes (beta)

## Data Availability

Project name: vvv2_display, Project home page: https://github.com/ANSES-Ploufragan/vvv2_display (accessed on 19 June 2025), Operating system (s): Platform independent, Programming language: python3, R, cheetah, Requirements: web browser, such as Firefox, Chrome or Safari, License: GNU license—GPL-3.0-only (GNU General Public License. version 3) (https://opensource.org/licenses/GPL-3.0 (accessed on 19 June 2025)), Any restrictions to use by non-academics: none, vvv2_display and vadr Galaxy wrappers are available on the Galaxy toolshed: https://toolshed.g2.bx.psu.edu/repository?repository_id=9cad962c7d2e7899 (accessed on 19 June 2025), Galaxy workflows are available on WorkflowHub: https://workflowhub.eu/workflows/1738 (accessed on 19 June 2025) (paired-end); https://workflowhub.eu/workflows/1739 (accessed on 19 June 2025) (single-end Illumina/Proton); https://workflowhub.eu/workflows/1740 (accessed on 19 June 2025) (Nanopore); https://workflowhub.eu/workflows/1741 (accessed on 19 June 2025) (PacBio); The Porcine Circovirus model database model for vadr is available at: https://zenodo.org/records/15065124 (accessed on 19 June 2025).

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
