# Peer review of "vvv2_align_SE, vvv2_align_PE/vvv2_display: Galaxy-Based Workflows and Tool Designed to Perform, Summarize and Visualize Variant Calling and Annotation in Viral Genome Assemblies"

_viruses, 2025, doi:10.3390/v17101385_

Round 1

Reviewer 1 Report

Comments and Suggestions for Authors

The manuscript by Alexandre Flageul and colleagues presents a new, user-friendly Galaxy-native visualization and summarization tool, vvv2_display, which consolidates viral variant calling, coverage, and functional annotations into two concise outputs: a genome-wide PNG plot and a cross-referenced TSV containing high-confidence variants. The authors integrate vvv2_display into Galaxy workflows covering Illumina and Nanopore/PacBio data and evaluated VADR and VAPiD annotators. The study includes a basic assessment of efficiency on a limited but important set of models, including several clinically relevant viruses, and demonstrates that the work addresses a practical gap for viral variant interpretation within Galaxy.

Major comments

  1. Manuscript structure. The discussion of tools is placed inside the Materials and Methods section. This discussion is important and interesting, but a little confusing there. I recommend restructuring the manuscript and moving it to another section.
  2. Including a compact comparative table explaining how vvv2_display differs from existing generic visualization utilities (inputs accepted, outputs produced, interactivity, Galaxy wrappers, maintenance status) would strengthen conclusions about necessity of the new tool.
  3. Please provide guidance by virus class (e.g., different DNA and RNA viruses) on selecting the threshold for variant proportion, with a brief sensitivity analysis.
  4. Results, Lines 175–182: the evaluation section lists multiple virus families and models, but results are largely qualitative. I recommend reporting numbers of features/genes predicted correctly vs. incorrectly, and perhaps concordance with reference annotations. Frankly speaking, the total number of tested sequences do not look very impressive, but is acceptable.
  5. Lines 87–107: parameters of softwares and tools used should be listed (at least general and most critical ones), so readers can reproduce the results.
  6. Lines 235–239: where you mention “less than 3 minutes” and “less than 13 GB RAM,” please specify the CPU model and thread count.

Minor comments

  1. Table 1 and elsewhere: unify capitalization of model/tool and virus names (e.g., “hepatitis C virus (HCV)”).
  2. As a practicing bioinformatician, I would recommend to describe the procedure of installation under conda environment. It would not take much place but it would improve the experience of use.

In general, the manuscript presents a useful, practice-oriented contribution that improves interpretability of viral variant data within Galaxy and is likely to be adopted by researchers. I recommend addressing the items above and adding compact quantitative benchmarks.

Reviewer 2 Report

Comments and Suggestions for Authors

This is potentially a useful tool to compare a reference virus genome with the query viral sample to identify any deviations. I think simpler tools like this are important for researchers and this tools appears to be worth publishing. 

I don't see any major flaws in the methodology and it is good that the authors have provided a github repo for local install of the program.

One thing that would substantially help the reader understand the workflow of the program is to add a flowchart in the Methods section to show the steps of the program. Right now with just the text, it is hard to visualize all the steps that the programs are performing.

Can the authors please do a more in depth discussion of how this tool compared to other tools? Perhaps include a table to summarize the differences between this tool and other similar tools for viral variant analysis?

Round 2

Reviewer 1 Report

Comments and Suggestions for Authors

The authors have thoughtfully addressed the reviewers’ feedback, leading to a significantly improved manuscript. The article is deemed appropriate for publication in the journal Viruses.